# Extracellular matrix analysis of fibrosis: A step towards tissue engineering for urethral stricture disease

**Emma C. Linssen[1], Jeroen Demmers[2], Christian G. M. van Dijk[3], Roos van Dam[1], Maria Novella Nicese[1], Caroline Cheng[3], Laetitia M. O. de Kort[1], Petra de Graaf[1]***

**1** Department of Urology, University Medical Center Utrecht, Utrecht, The Netherlands, **2** Department of Proteomics, Erasmus Medical Center, Rotterdam, The Netherlands, **3** Department of Nephrology, University Medical Center Utrecht, Utrecht, The Netherlands

\* p.degraaf-4@umcutrecht.nl

**Data Availability Statement:** All relevant data are within the manuscript and its Supporting Information files.

## Abstract

The urogenital tract is a target for many congenital and acquired diseases, both benign and oncogenic. In males, the urethra that transports urine and semen can be obstructed by a fibrotic disease called urethral stricture disease (USD). In severe USD, the whole organ including the vascular embedding, the corpus spongiosum (CS), is affected. Recurrent or severe USD is treated by reconstructive surgery. Tissue engineering may improve the outcome of urethral reconstruction in patients with complicated USD. Currently in urethral reconstruction only the epithelial layer is replaced, no substitution for the CS is provided, while the CS is important for mechanical support and vascularization. To develop a tissue engineering strategy for the CS, it is necessary to know the protein composition of the CS. As the extracellular matrix (ECM) plays an important role in the formation of fibrosis, we analyzed the distribution and localization of ECM components in human healthy and fibrotic CS tissue using immunohistology. The morphology of components of the elastic network were affected in USD. After decellularization a clear enrichment of proteins belonging to the ECM was found. In the proteomic analysis collagens COL15A1 and COL4A2 as well as inter-alpha-trypsin inhibitor ITIH4 were upregulated in fibrotic samples. The glycoproteins Periostin (POSTN), Microfibrillar-associated protein 5 (MFAP5) and EMILIN2 are downregulated in fibrotic tissue. To our knowledge this is the first proteomic study of ECM proteins of the CS in healthy and in USD. With these results a regenerating approach for tissue engineered CS can be developed, including relevant ECM proteins that reduce fibrosis and promote healthy healing in urethral reconstructive surgery.

## Introduction

Diseases of the urethra affect millions of men around the world. There are various pathologies that have different prevalence and severity. One of the most common afflictions of the urethra is urethral stricture disease (USD), which presents as a fibrous obstruction of the urethra hindering proper micturition and ejaculation. As in other soft tissues, this fibrosis often develops during a

**Funding:** The author(s) received no specific funding for this work.

wound-healing process triggered by an injury, which can be iatrogenic, traumatic, or inflammatory in origin, or the stricture may be congenital or idiopathic. Surgical treatment consists of excision of the fibrotic part and primary anastomosis (EPA) or grafting when the stricture is long, using buccal mucosa or prepuce. The corpus spongiosum (CS) is an integral part of the urethra [1], provides support and is essential for healing of the urethra [2]. In case of advanced USD, the fibrotic process extends from the epithelium into the CS, so called spongiofibrosis.

In urethral reconstruction only the mucosa of the urethra is replaced, while the vascular and mechanical support of the urethra, the CS, is not provided. Many complications associated with urethral reconstruction are due to inadequate vascularization [3]. In contrast to the mucosa of the urethra, there is no autologous tissue that can be used as a substitution for CS. The focus of our research is tissue engineering strategies in USD. With a combination of cells, (bio)materials and biochemical signals the goal is to develop a graft to be used to repair or to replace diseased tissue, including strictures in the urethra. To overcome complications in urethral reconstruction, tissue engineering of both the urethral mucosa and the CS could be a solution. The CS is composed of several layers [4], which could be mimicked using 3D-bioprinting. Hydrogels are currently used for cell loaded bioprinting. Existing hydrogels like Gelatin Methacryloyl (GelMA) have good mechanical properties for printing [5], but lack the required biochemical cues for cells.

To recreate CS that mimics healthy native vascular structures, the exact structure and composition of healthy native CS should be known. The extracellular matrix (ECM) is the essential non-cellular component of the tissue microenvironment of cells, comprised of a network of macromolecules including polysaccharide glycosaminoglycans (GAGs) and proteins including collagens, laminins, and fibronectin [6]. Previously, studies to the macro and micro architecture have shown that a large part of the structure of the CS consists of collagens, elastin and other ECM components [4, 7]. It is known that USD affects the ECM of CS [8, 9].

By elucidating the differences in distribution and expression of ECM proteins in fibrotic and healthy CS, a regenerating approach can be developed with relevant ECM proteins that reduce fibrosis and promote healthy healing. The aim of this study was to identify ECM components in both healthy and fibrotic CS. We focus on the proteomic composition of the ECM to create a better insight of ECM proteins present in the CS.

## Materials and methods

### Human tissue

After informed consent, obstructed urethra (surgical waste material) was collected from three patients during excision and primary anastomosis (EPA) surgery under local biobank protocol (UMC Utrecht, Medisch Ethische Toetsingscommissie (METC) 15–393). Only adult patients were included and written consent was given before the procedure. All patients had a history of at least one endo-urethral treatment (direct vision internal urethrotomy or endoscopic dilatation). The cause of the primary stricture (traumatic, inflammatory, iatrogenic) was unknown. Tissue was processed by fixation in formaldehyde and subsequent embedding in paraffin for (immune)histology or by freezing and storage at -80°C for decellularization and subsequent proteomic analysis.

As control tissues three male penile spongy urethras from gender conformation surgery were used [4]. Tissue samples were taken from the penile urethra (mid-shaft and distal).

### Histology and immunohistology

Tissue was embedded in paraffin and sections of 3 μm were made using a Leica microme microtome.

**DNA content.** Tissue sections on slides were deparaffinized using series of histoclear, 70% ethanol and 100% ethanol, washed with PBST (phosphate buffered saline + 0.1% w/v Triton X-100), permeabilized for 20 min with 0.5% Triton-X100 in phosphate buffered saline (PBS) followed by rinsing in PBS twice, incubated 5 min with 4′,6-diamidino-2-phenylindole (DAPI), washed 2x with demiwater, dehydrated using series of 100% ethanol, 70% ethanol and histoclear, and mounted in mowiol.

**(Immuno)Histochemistry.** For Haematoxylin-Eosin staining, tissue sections on slides were deparaffinised, washed twice with demiwater, stained for 5 min with Mayer's haematoxylin, developed for 10 min in tap water, stained for 30 seconds with eosine, washed with demiwater until clear, dehydrated using series of 100% ethanol, 70% ethanol and histoclear, and mounted in entellan. For morphological analysis 3 μm paraffin sections were stained Masson's trichrome and Sirius red.

For Immunohistochemistry, tissue sections on slides were deparaffinized, blocked using 5% $H_2O_2$ in PBS followed by antigen retrieval using citrate-buffer (10 mM sodium-citrate, pH 6.0) for 20 min at 95˚C. After cooling, slides were washed 3x with PBST and blocked for 30 min with PBS/2%BSA (Bovine serum albumin, w/v). Primary antibody was incubated for an hour at room temperature (rT) or overnight at 4˚C. Slides were washed 3x with PBST and incubated with the secondary antibody for 30–45 min at rT. In some cases, a third antibody was used for enhancement of the signal (see S1 Table), for 45 min at rT. The second and third antibody was coupled to horse radish peroxidase (HRP). Slides were washed 3x with PBS and incubated for 8 min with Novared peroxidase substrate (Vector Laboratories, SK-4800), counterstained with haematoxylin and dehydrated followed by mounting in entellan. Overview of the antibodies used is given in S1 Table.

## Decellularization

Several different procedures were performed to isolate the ECM of the frozen urethral samples. First, Triton-X100 decellularization at 4˚C was performed, in short: Approximately 1 cm of spongy urethra (transgender or fibrotic tissue) was placed in 40 ml $H_2O$ for 24h and subsequently in Triton-X100 solution (1% Triton, 0.1% $NH_4OH$) for 2x 24h and 3x 72h, followed by 48h in $H_2O$.

Alternatively, sodium dodecyl sulphate (SDS) was used at rT, as follows: Approximately 1 cm of spongy urethra (transgender or fibrotic tissue) was placed in 40 ml $H_2O$ for 45 min and subsequently in SDS-solution (1% SDS in potassium-free PBS) for 4x 24h and 1x 72h followed by 2x 24h in $H_2O$. All steps were performed at rT to prevent crystallization of SDS. Microdissection was applied to transgender as well as fibrotic tissue to isolate CS as good as possible. Typical example of fibrotic tissue and healthy tissue before and after decellularization is given in S1 Fig. After decellularization tissue was cut into small pieces (1x1x1 mm) and stored in PBS before processing to LC-MS/MS analysis.

## LC-MS/MS analysis

Decellularized surgical waste samples were processed as described before [10, 11]. In short, SDS-PAGE-separated samples were prepared and proteins lanes were cut out of the gel, reduced with dithiothreitol, alkylated with iodoacetamide, and digested with trypsin, as described previously [12]. Supernatants were stored in glass vials at −20˚C until further measurements. An EASY-nLC system (Thermo, The Netherlands) was used for nanoflow LC-MS/MS coupled to a Q Exactive Plus mass spectrometer (Thermo, The Netherlands) operating in positive mode and equipped with a nanospray source, as previously described [13]. Samples were trapped on a 1.5 cm × 100 μm in-house packed ReproSil C18 reversed phase column (Dr

Maisch GmbH, Ammerbuch-Entringen, Germany) at a flow rate of 8 μL/min. Sequentially, samples were separated on a 15 cm × 50 μm in-house packed ReproSil C18 reversed phase column (Dr Maisch GmbH) by adding a linear gradient from 0%–80% solvent B in solvent A, where A consisted of 0.1% formic acid and B of 80% (v/v) acetonitrile and 0.1% formic acid. Flow rate was set at 200 nL/min, and elution took place over 70 min. The eluent was sprayed by a nanospray device directly into the ESI source of the LTQ ion trap mass spectrometer. Mass spectra were acquired in continuum mode, and peptide fragmentation was performed in data-dependent mode. MS/MS spectra were extracted out of raw data files and were analysed by using MaxQuant software (version 2.1.4.0), as previously described [14] and MaxQuant results were analysed using Perseus (version 1.6.0.7). Proteins that were identified with at least two values in one group (healthy, fibrotic) were taken along in the analysis. Proteins that were only identified by site, reverse and potential contaminants were removed. NaN values were imputed by random values taken from a normal distribution of background signal. LFQ intensity was used for quantification. Proteins with a log 2 fold expression difference or more between groups were selected as proteins of interest in a volcano plot, as well as proteins that proved significant ($p < 0.05$) in a t-test. By cross referencing with the Human Matrisome Project [15] the results were analysed using GO biological process/molecular function/cellular component analysis (geneontology.org) and STRING network analysis (string-db.org).

## Results

### ECM in healthy CS and in USD

Differentially distributed connective tissue (blue fibres) was shown in a Masson Trichrome staining of healthy and fibrotic CS (Fig 1A and 1B). Note the difference in epithelium: a multi-layered epithelium (cytoplasmic keratins stain pink) in the urethra was observed in the fibrotic tissue (Fig 1B) combined with a severely reduced lumen compared to the healthy urethra where a single layer of epithelial cells was shown (Fig 1A). Sirius red staining demonstrate that the distribution of the collagen components was more evenly spread in the healthy tissue compared to the diseased (Fig 1C and 1D). No significant differences in total collagens were found when comparing the signal intensity for fibrotic to healthy tissue (results not shown). Next the tissues were stained for collagen type 1, 3 and 4 (Fig 2). In the healthy tissue collagen fibres were more organized (Fig 2A, 2C and 2E) compared to the fibrotic tissue (Fig 2B, 2D and 2F). Less signal was seen in fibrotic tissue for COL3 (compare 2C to 2D).

### SDS based decellularization conserved the major ECM components

Comparison in decellularization procedure is depicted in S1 Fig. Both Triton-X100 (S1A–S1D Fig) and SDS (S1E–S1H Fig) shows retention of the tissue structure. Cell nuclei are stained purple (in HE) respectively blue in DAPI. Although for both methods nuclei were detected in dense structures, more nuclei were detected in tissue decellularized by Triton-X100, in healthy tissue as well as in the–more impenetrable- fibrotic tissue.

Proteome composition analysis of decellularized samples by LC-MS/MS showed much more enrichment of the matrisome by SDS decellularization compared to Triton-X100 decellularization. Especially core matrisome components as collagens, laminins and proteoglycans were removed during Triton X-100 decellularization (S2 and S3 Tables), while these were retained using SDS.

Based on these results SDS was chosen as best for decellularization method and further analysis.

Masson Trichrome showed that connective tissue retained, but cytoplasmic components were no longer detected after SDS decellularization (compare Fig 3A and 3B with Fig 1A and 1B).

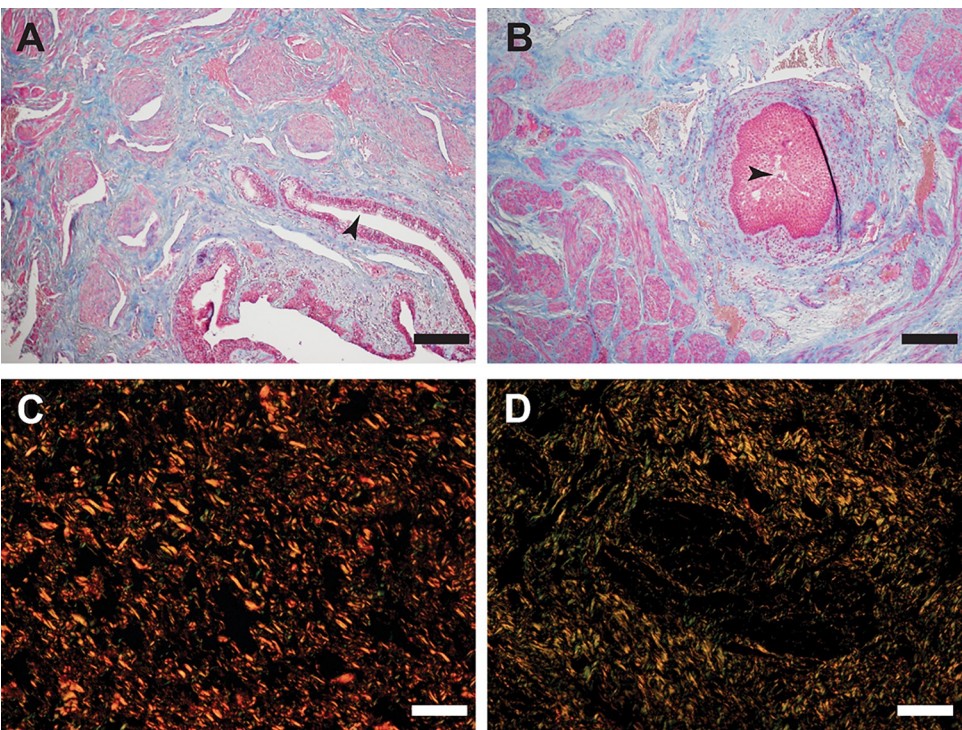

**Fig 1. Histology.** A) Healthy, Masson Trichrome (MT). B) Fibrosis MT. C) Healthy, Picrosirius Red (PR). D) Fibrosis PR. A, B: scale bar represents 200 μm, C and D scale bar represents 40 μm. Arrowheads point out the epithelium in A and B.

Collagen fibres were still present, with more fibres detected in fibrotic tissue (compare blue staining in Fig 3A with 3B). Total collagens showed no difference with non-decellularized tissue in signal analysis after Sirius Red staining (compare Fig 3C and 3D with D compared to Fig 1C and 1D).

## Comparison ECM composition in healthy and fibrotic samples

In total 1154 unique proteins were detected in decellularized tissue from healthy and fibrotic samples in the proteomic analysis, of which 236 proteins are associated to the matrisome [15]. Most matrisome proteins were detected in more than one sample. ECM distribution pie charts show not much variation in the total amount of ECM proteins detected in both healthy and fibrotic tissue (Fig 4A and 4B). The majority of matrisome proteins that are more or less abundant present in either healthy or fibrotic tissue are glycoprotein (red) or collagen (green) derived (Fig 4C). Despite cellular and cytoskeletal proteins being detected, decellularization of the tissue enriched the ECM proteins composition in the proteomic analysis since 86% of the total protein amount is ECM derived (Table 1). A full overview of the iBAQ and LFQ values of all detected proteins is given in the excel file in the supplemental data deposited together with this manuscript.

In total amount the collagens were most abundant in our analysis (Table 1) followed by the glycoproteins. Of the collagens, COL1, COL3, COL4 and COL6 were highest detected, except for COL4A2 (see below) no significant difference was detected between healthy and fibrotic samples nor in the COL1/COL3 ratio (Table 2 and S3 Fig).

Collagen types COL15A1 and COL4A2 were upregulated in fibrotic tissue, as well as the matrisome associated inter-alpha-trypsin inhibitor ITIH4. The glycoproteins Periostin (POSTN), Microfibrillar-associated protein 5 (MFAP5) and Elastin microfibril interfacer 2

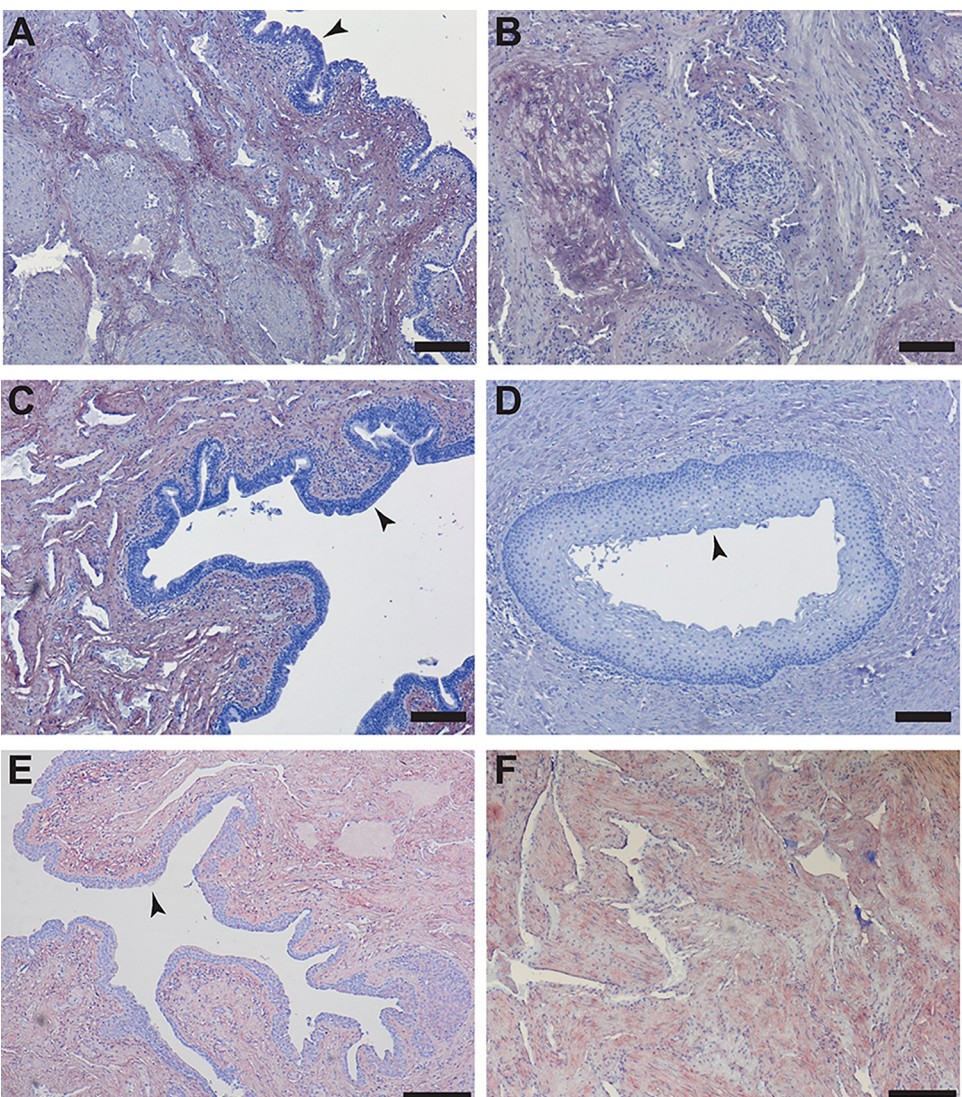

**Fig 2.** IHC staining for Collagens A) Healthy, Collagen (COL) 1. B) Fibrosis COL1. C) Healthy, COL3. D) Fibrosis COL3. E) Healthy, Collagen (COL) 3. F) Fibrosis COL3. Panels A, B, C, D, E, F: scale bar represents 200 μm. Arrowheads point out the epithelium in A, C, D, E.

(EMILIN2) were downregulated in fibrotic tissue (Fig 5A–5F). Fibrillin 2 (FBN2) was also downregulated, however, due to large variation not significant (S3 Fig). Overview of the LFQ values for these proteins is given in Table 3.

Various elastin (ELN) peptides were detected in all samples and these could be mapped to several different isoforms. Some of these peptides were differentially detected among the samples, however, careful comparison did not indicate a differential expression of one or more of the isoforms (results not shown).

## Validation of ECM in immunohistochemistry

As the glycoproteins MFAP5, FBN2 and EMILIN2, belonging to the elastin—microfibril network, were downregulated, we decided to validate the findings by IHC for the more highly

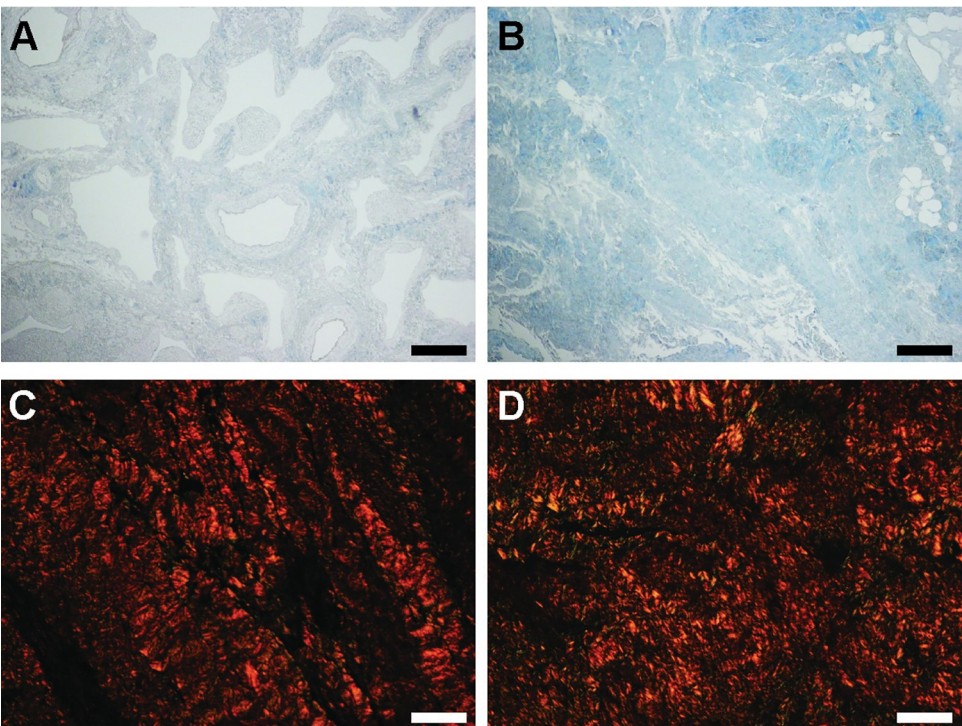

**Fig 3. Histology ECM components.** A) Decellularized healthy tissue, stained with Masson Trichrome (MT). B) Decellularized fibrotic tissue, stained with MT. C) decellularized healthy tissue, stained with Picrosirius Red (PR). D) Decellularized fibrotic tissue, stained with PR. A-B scale bar represents 200 μm, panels C-D scale bar represents 40 μm.

expressed ELN and the isoforms FBN1 and EMILIN1 all belonging to the same network. LFQ values are given in Table 3, ratios between the isoforms in Table 4. ELN was detected in bundles in healthy tissue and decellularization had no effect on this distribution (Fig 6A and 6C). In the fibrotic tissue elastin was less organized (Fig 6B and 6D). No loss of ELN staining was found after decellularization (compare Fig 6A to 6C and Fig 6B to 6D). A staining for elastic ECM components EMILIN1 and FBN1 followed the same pattern as shown for elastin (Fig 6E–6G). The intensity seemed lower in the fibrotic tissue compared to healthy, however, quantification of the signal could not confirm this (results not shown).

## Discussion

In this study we characterized the ECM proteins present in both healthy and fibrotic CS tissue. Distribution of ECM proteins like collagens and glycoproteins of the microfibrils/elastic fibre network were affected by USD, which is in line with previous findings that the ECM is structurally altered during disease, especially as a result of chronic inflammation and fibrosis [6]. Overall protein intensities of ECM components were comparable in the proteomic analysis of healthy and diseased tissue.

While other studies report a shift in COL1/COL3 ratio form 3:1 in healthy CS and 5.2:1 in diseased [16], we found no significant shift between healthy and fibrotic CS. Although in immunohistochemistry the expression of COL3 seemed lower, this was not confirmed in the proteomic analysis. The difference in collagen ratio could be explained by the different methods that were used. Where we use LC-MS/MS based proteome analysis of decellularized tissue, Baskin *et al* used hydroxyproline analysis of intact tissue including all cellular components

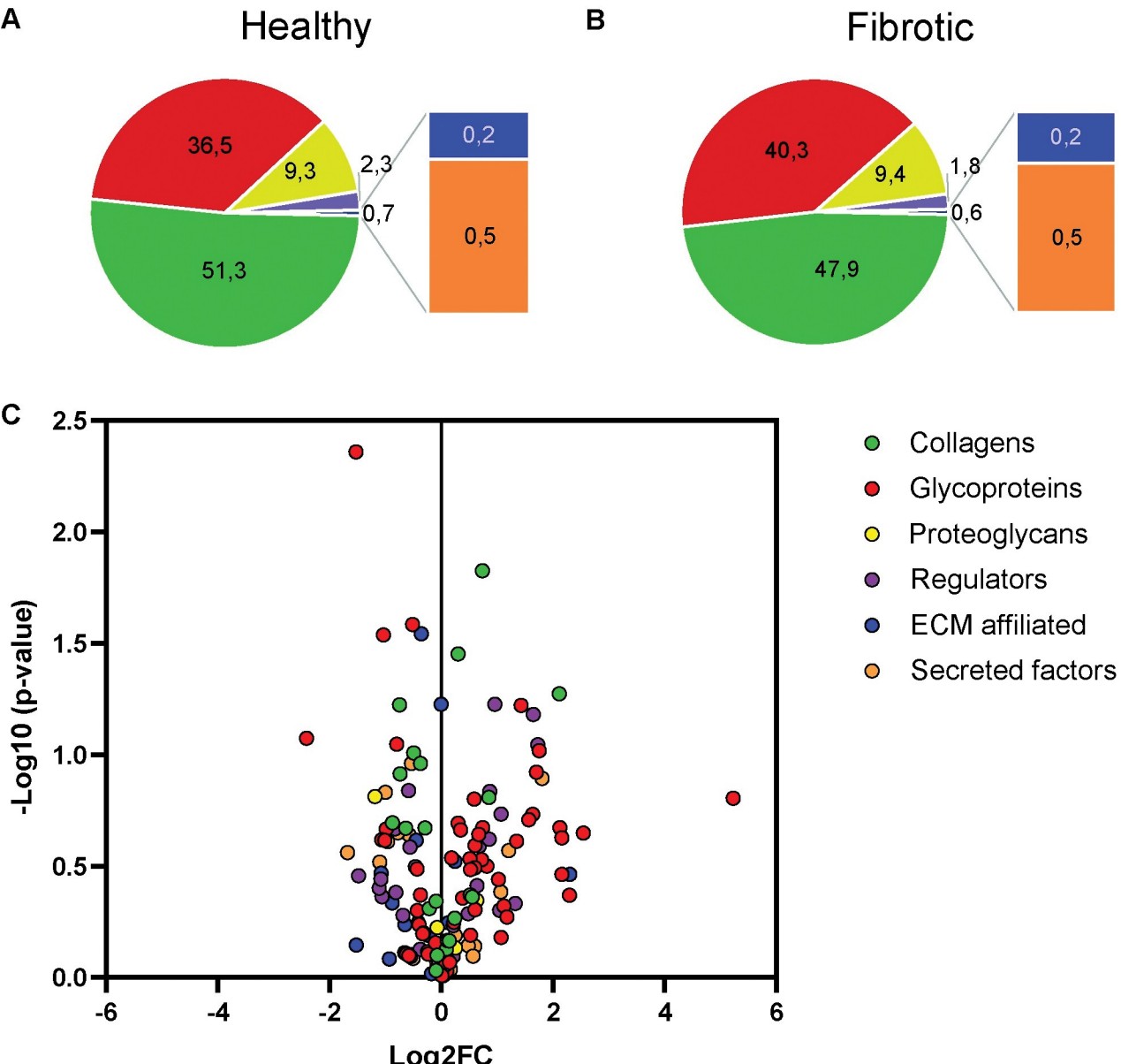

**Fig 4. Pie Charts and Volcano plot of LC-MS/MS results.** A) Distribution of matrisome proteins measured with LC-MS/MS in healthy tissue. B) Distribution of matrisome proteins measured with LC-MS/MS in fibrotic tissue. C) Log2 fold change over healthy ECM (x-axes) and significance (y-axes). In green the collagens, in red the glycoproteins, yellow the proteoglycans, in purple the ECM regulators, in blue the ECM affiliated proteins and lastly in orange the secreted factors. Overview of the fold change in expression of these proteins is given in **S4 Table**.

[16]. The advantage of our approach is the unbiased character of proteomics. Not only presence or absence of a particular protein itself, but also the ratios between the proteins need to be taken into account in developing a regenerative approach to healthy healing in USD, proteomics can give that in a quantitative fashion.

When comparing healthy CS with fibrotic CS, the proteins POSTN, MFAP5, EMILIN2 and FBN2 were more highly expressed in healthy tissue. Elastin (ELN) was found in all samples and there was no difference in average expression. Previous reports on fibrotic tissue from urethral strictures showed more elastin fibres in the fibrotic lesion proximal to the urethral

**Table 1. Matrisome protein groups as % of total protein detected.**

|  | H1 (%) | H2 (%) | H3 (%) | SF1 (%) | SF2 (%) | SF3 (%) | Av H ± s.d. | Av SF± s.d. |
|---|---|---|---|---|---|---|---|---|
| Total matrisome | 88.4 | 88.8 | 83.5 | 87.8 | 88.9 | 82.5 | 86.9 ± 3.4 | 86.4 ± 3.0 |
| Collagens | 48.74 | 40.64 | 44.33 | 46.11 | 40.89 | 37.77 | 44.57 ± 4.06 | 41.59 ± 4.21 |
| Glycoproteins | 30.42 | 35.88 | 28.74 | 30.91 | 36.42 | 36.15 | 31.68 ± 3.7 | 34.50 ± 3.1 |
| Proteoglycans | 8.07 | 8.67 | 7.46 | 8.28 | 9.30 | 6.84 | 8.07 ± 0.60 | 8.14 ± 1.24 |
| Regulators | 0.80 | 2.88 | 2.22 | 2.01 | 1.58 | 1.28 | 1.97 ± 0.60 | 1.62 ± 0.60 |
| ECM Affiliated | 0.11 | 0.15 | 0.16 | 0.14 | 0.16 | 0.13 | 0.14 ± 0.03 | 0.14 ± 0.02 |
| Secreted factors | 0.21 | 0.57 | 0.57 | 0.39 | 0.51 | 0.34 | 0.45 ± 0.21 | 0.41 ± 0.09 |

obstruction [8] compared to distal. We did not make this distinction between proximal and distal to the obstruction, but analysed the fibrotic lesion as a whole. In healthy elastic tissue, ECM components are balanced in expression. We found a completely different distribution of ELN in fibrotic tissue in IHC, in combination with less peptides detected for MFAP5, EMI-LIN2 and FBN2 in the proteomic analysis.

Most proteins more highly expressed in healthy, and downregulated in fibrotic tissue, are all involved in the formation of the microfibrils/elastic fibre network. The microfibrillar-associated protein 5 (MFAP5) interacts with Fibrilin1 and 2 (FBN1 and FBN2). Loss of function mutation of MFAP5 leads to familial thoracic aortic aneurysms and dissections [17], indicating a role in the ECM of vascular tissue. FBN2 containing microfibrils regulate the early process of elastic fibre assembly to the microfibrils/elastic fibre network [18]. EMILIN2 deposition and network formation depends on other ECM networks, such as fibrillin microfibrils [19]. By

**Table 2. Collagen isoforms as % of total protein detected.**

| collagen isoforms | H1 | H2 | H3 | SF1 | SF2 | SF3 | Average H | Average SF | Fold change (SF/H) | p value |
|---|---|---|---|---|---|---|---|---|---|---|
| COL1A1 | 14.99 | 10.31 | 16.39 | 9.42 | 10.55 | 9.68 | 13.89 | 9.88 | 0.71 | 1.01 |
| COL1A2 | 8.59 | 7.72 | 10.84 | 6.96 | 6.34 | 7.66 | 9.05 | 6.99 | 0.77 | 0.96 |
| COL3A1 | 1.31 | 1.47 | 2.30 | 0.79 | 0.94 | 1.32 | 1.69 | 1.02 | 0.60 | 0.91 |
| COL4A1 | 3.77 | 3.19 | 3.70 | 2.93 | 3.47 | 3.58 | 3.55 | 3.33 | 0.94 | 0.34 |
| COL4A2 | 3.27 | 3.38 | 2.68 | 3.73 | 3.79 | 3.99 | 3.11 | 3.84 | 1.23 | 1.45 |
| COL4A3 | 0.00 | 0.00 | 0.00 | 0.00 | 0.00 | 0.00 | 0.00 | 0.00 | # | 0.93 |
| COL4A5 | 0.24 | 0.17 | 0.17 | 0.15 | 0.17 | 0.16 | 0.19 | 0.16 | 0.82 | 0.67 |
| COL4A6 | 0.23 | 0.19 | 0.12 | 0.16 | 0.19 | 0.17 | 0.18 | 0.17 | 0.95 | 0.10 |
| COL5A1 | 0.05 | 0.04 | 0.08 | 0.02 | 0.03 | 0.05 | 0.06 | 0.04 | 0.64 | 0.67 |
| COL5A2 | 0.19 | 0.17 | 0.14 | 0.15 | 0.16 | 0.23 | 0.17 | 0.18 | 1.08 | 0.17 |
| COL6A1 | 5.21 | 4.43 | 2.05 | 5.20 | 4.92 | 3.69 | 3.90 | 4.60 | 1.18 | 0.27 |
| COL6A2 | 2.95 | 2.51 | 1.24 | 2.82 | 2.46 | 2.13 | 2.23 | 2.47 | 1.11 | 0.16 |
| COL6A3 | 6.73 | 5.84 | 3.64 | 12.42 | 6.71 | 4.06 | 5.40 | 7.73 | 1.43 | 0.37 |
| COL8A1 | 0.00 | 0.06 | 0.06 | 0.06 | 0.07 | 0.04 | 0.04 | 0.06 | 1.47 | 0.36 |
| COL8A2 | 0.00 | 0.00 | 0.00 | 0.00 | 0.00 | 0.00 | 0.00 | 0.00 | 0.94 | 0.03 |
| COL12A1 | 0.07 | 0.07 | 0.04 | 0.10 | 0.16 | 0.07 | 0.06 | 0.11 | 1.81 | 0.81 |
| COL14A1 | 0.89 | 0.86 | 0.53 | 1.00 | 0.72 | 0.70 | 0.76 | 0.81 | 1.07 | 0.13 |
| COL15A1 | 0.02 | 0.02 | 0.01 | 0.03 | 0.03 | 0.03 | 0.02 | 0.03 | 1.66 | 1.83 |
| COL16A1 | 0.09 | 0.10 | 0.21 | 0.07 | 0.06 | 0.09 | 0.13 | 0.07 | 0.55 | 0.69 |
| COL18A1 | 0.04 | 0.04 | 0.03 | 0.03 | 0.05 | 0.03 | 0.04 | 0.04 | 0.86 | 0.31 |
| COL21A1 | 0.09 | 0.07 | 0.09 | 0.04 | 0.04 | 0.07 | 0.08 | 0.05 | 0.60 | 1.22 |
| COL28A1 | 0.00 | 0.01 | 0.00 | 0.01 | 0.01 | 0.01 | 0.00 | 0.01 | 4.33 | 1.27 |

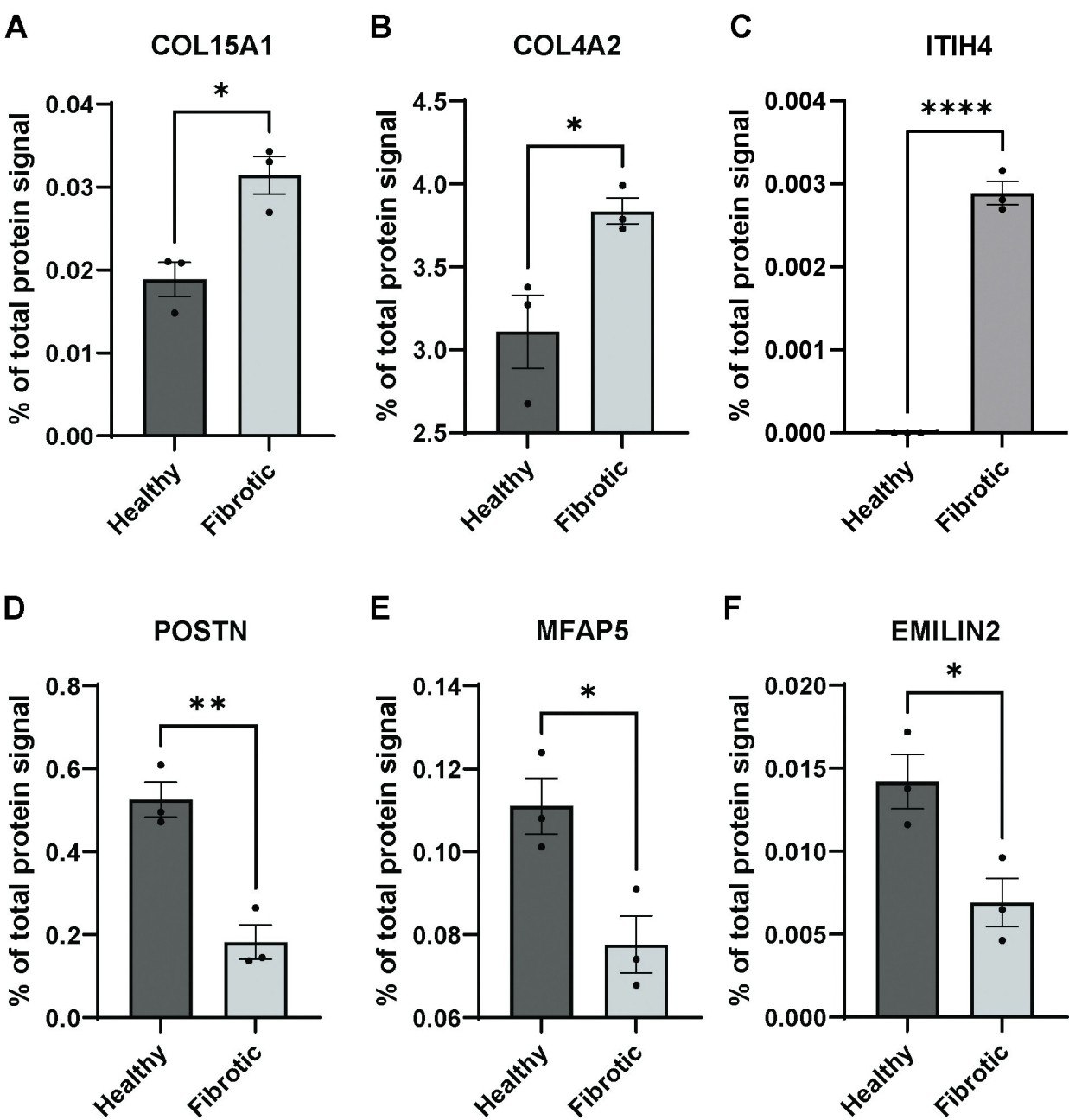

**Fig 5. Overview of up and downregulated proteins in fibrotic tissue.** Dark grey bars represent healthy tissue, light grey bars fibrotic tissue. Y-axes represent percentage of total protein detected in the decellularized tissue. * $p < 0.05$, ** $p < 0.005$ and **** $p < 0.0001$.

interaction the elastin and microfibrils form a supramolecular scaffold [18]. This scaffold not only provides the right biomechanical properties to the tissue, but also regulates the availability of essential growth factors. As elastin, EMILIN1/2, MFAP5 and FBN1/2 are all in the microfibrillar network, the interaction of the proteins and the ratio may be more important than the absolute amounts.

POSTN was also more highly expressed in healthy tissue compared to fibrotic. It has been shown that pericytes produce POSTN and thereby inducing angiogenesis [20]. In addition,

**Table 3. LFQ values for specific proteins.**

|  | H1 | H2 | H3 | SF1 | SF2 | SF3 |
|---|---|---|---|---|---|---|
| COL15A1 | 1.93E+08 | 1.91E+08 | 1.31E+08 | 2.18E+08 | 2.99E+08 | 3.54E+08 |
| COL4A2 | 3.03E+10 | 3.07E+10 | 2.37E+10 | 3.01E+10 | 3.43E+10 | 4.11E+10 |
| ITIH4 | 0 | 0 | 0 | 2.27E+07 | 2.86E+07 | 2.78E+07 |
| POSTN | 5.64E+09 | 4.29E+09 | 4.39E+09 | 2.14E+09 | 1.24E+09 | 1.50E+09 |
| MFAP5 | 9.37E+08 | 1.13E+09 | 9.57E+08 | 5.98E+08 | 6.14E+08 | 9.38E+08 |
| ELN | 7.04E+09 | 3.46E+10 | 2.10E+10 | 1.00E+10 | 1.62E+10 | 2.29E+10 |
| EMILIN1 | 8.17E+09 | 6.72E+09 | 5.74E+09 | 9.38E+09 | 8.95E+09 | 7.59E+09 |
| EMILIN2 | 1.28E+08 | 1.05E+08 | 1.52E+08 | 7.77E+07 | 5.87E+07 | 4.77E+07 |
| FBN1 | 1.52E+11 | 1.62E+11 | 1.27E+11 | 1.24E+11 | 1.14E+11 | 2.18E+11 |
| FBN2 | 8.10E+08 | 3.42E+08 | 1.26E+09 | 2.63E+08 | 1.59E+08 | 0 |

POSTN is an important structural mediator, involved in tissue adaption in response to insult or injury [21]. Skin regeneration in response to insult is associated with increased POSTN expression [22]. However, this phenomenon is only transient, starting a few days post-injury, with protein levels peaking around a week after insult followed by a decreased expression. When comparing skin wound healing with oral mucosal wound healing, expression is higher in healed skin compared to healed oral wounds [23], whereas in skin healing more scars are formed compared to oral tissue. This seems in contrast with our findings: we found POSTN more highly expressed in healthy tissue compared to the scarred, fibrotic urethra. On the other hand, expression could have been decreased as USD is not an acute disease. Spongiofibrosis is the end result of defective healing, the wound healing phase is already over. Another reason could be loss of pericytes in spongiofibrosis: healthy penile tissue is rich in pericytes [24] and in the loss of vascular tissue including supporting pericytes, POSTN expression could be reduced.

In the proteomic analysis collagens COL15A1 and COL4A2 as well as ITIH4 were upregulated in fibrotic samples. These proteins have been implemented in fibrosis before, however, not in USD. COL4A2 is upregulated in both liver and lung fibrosis [25, 26]. Less is known on the role of COL15A1 in fibrosis, however, an upregulation is found in small airway fibrosis [27]. ITIH4 is a protease inhibitor involved in inflammation and was found as a biomarker for liver fibrosis [28, 29]. Given their indicated roles in fibrotic disease in other organs or tissues, upregulation in USD can be explained by the fact that fibrosis in several organs is caused by a similar pathway [30].

All matrisome proteins identified have not been associated with USD previously. Multiple factors from the microfibrillar network were disturbed in USD. This may indicate that the response after injury of the urethra/CS is followed by fast and defective healing to protect surrounding tissue from urine exposure, leading to a malfunctional microfibrils/elastic fibre network and eventually to fibrosis. We found that the overall ECM protein distribution was not altered in fibrotic tissue. However, some components of the microfibril network (FBN2, EMILIN2, MFAP5 and POSTN) are downregulated in fibrosis. These proteins should be included in regenerative strategies such as enrichment of hydrogels with ECM components [31] as they

**Table 4. Ratio's isoforms.**

|  | H1 | H2 | H3 | SF1 | SF2 | SF3 | Average H | Average SF |
|---|---|---|---|---|---|---|---|---|
| COL1/COL3 | 17.97 | 12.29 | 11.82 | 20.71 | 17.94 | 13.14 | 13.54 | 16.58 |
| EMILIN1/EMILIN2 | 64.0 | 63.8 | 37.7 | 121 | 152 | 159 | 55.2 | 144 |
| FBN1/FBN2 | 188 | 473 | 100 | 471 | 716 | # | 254 | # |

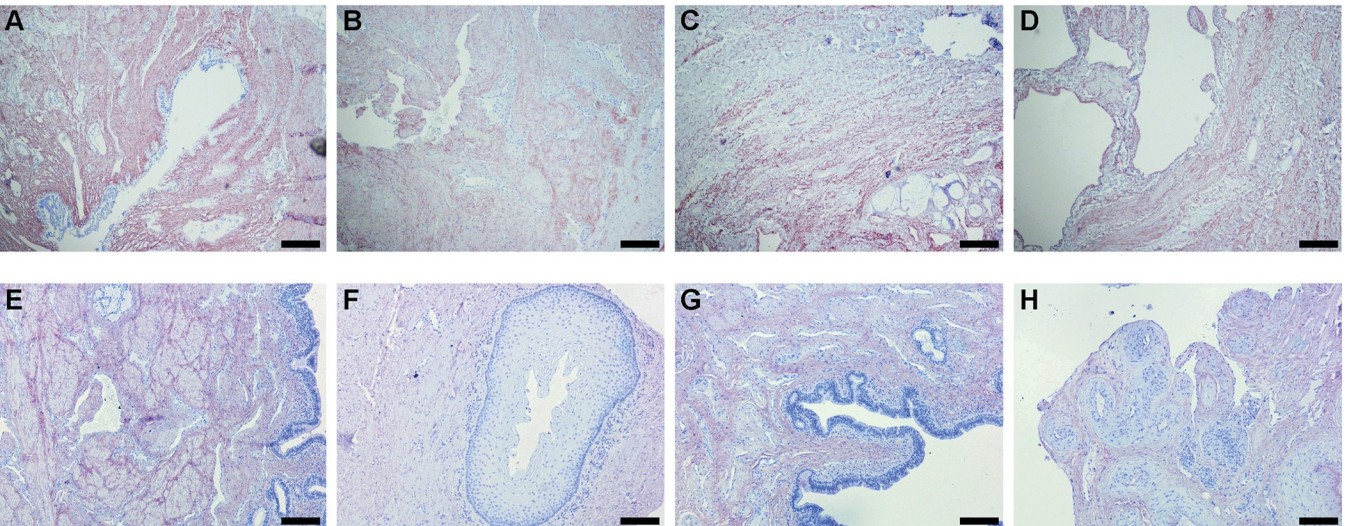

**Fig 6. Histology staining of glycoproteins.** A) Healthy, Elastin. B) Fibrosis, Elastin. C) Healthy, decellularized, Elastin. D) Fibrosis decellularized, Elastin. E) Healthy, EMILIN1. F) Fibrosis, EMILIN1. G) Healthy, FBN1. H) Fibrosis, FBN1. Scale bar represents 200 μm.

seems to play a role in the formation of the elastic network with ELN. We are not sure whether the proteins itself are important, or that the biomechanical properties, especially the elasticity, of the graft would play a role in healthy healing. For the same reason, exclusion of the collagens COL4A2 and COL15A1 and the IHIT4 protein is preferred in generating an ECM enriched hydrogel for 3D printing.

A limitation to this study is the use of surgical waste from gender conformation surgery: it is unclear to what extent the tissue represents actual healthy tissue. All our tissues originate from anonymous biobanks so the exact medical history is unknown. Trans females are generally treated with feminising hormones before gender confirming surgery [32]. It is not known how these altered hormone levels influence the CS, but there is evidence that hormones can affect ECM proteins [33, 34]. Transgender surgical waste was the best material available, because obtaining healthy human urethral tissue is very difficult. Human spongy urethra from potent young men is not readily available. Any manipulation or biopsy of the urethra may induce stricture disease. Initial experiments using cadaveric samples show degenerating of spongious tissue due to androgen depletion and erectile dysfunction because of old age (results not shown).

Animal specimens are not a valid alternative option as humans have a unique penile anatomy [35]. Another limitation is that we analysed FBN1 and EMILIN1 while our results showed differences in FBN2 and EMILIN2. We found the expression levels of the FBN1 and EMILIN1 40–700 fold higher than of the 2-isoforms (Table 4) and as the isoforms are similar in function and in interactions, we choose to analyse the 1-isoforms. Lastly, cellular proteins were found in the decellularized tissue, indicating sub-optimal decellularization. However, the proposed protocol was sufficient to enrich our proteomic analysis with ECM proteins. Lastly, the modular and repetitive nature of ELN may not support reliable detection of the protein by mass spectrometry. However, the results of the elastin-associated proteins showed a different distribution of the microfibril network in USD.

## Conclusion

To our knowledge this is the first proteomic study of ECM proteins in the spongious tissue of both healthy urethra and urethral stricture disease. We found that the overall protein

intensities were comparable, however, several ECM components were differently expressed in healthy CS compared with the CS in USD. A change in the distribution of the elastic fibres was observed while the total amount of elastin was comparable between healthy and fibrotic CS. Although the exact role of the here identified proteins needs to be elucidated, our findings may contribute to the use of ECM proteins in promoting the regeneration of healthy, non-fibrotic CS in the reconstructive treatment of USD.

## Supporting information

**S1 Fig. Typical example of healthy and fibrotic tissue before and after decellularization.** A) Fibrotic tissue before decellularization. B) Fibrotic tissue after decellularization (SDS). C) Healthy tissue before decellularization. D) Healthy tissue after decellularization (SDS).
(PDF)

**S2 Fig. Comparison of decellularization methods.** A-D) Triton-X100. E-G) SDS. A, C, E, G, Healthy tissue, B, D, G, H Fibrotic tissue. A, B, E, F: HE-staining. C, D, G, H: DAPI-staining. Scale bar represents 200 μm, except in B and D, scale bar represents 40 μm.
(PDF)

**S3 Fig. Comparisons of protein expression and COL1/COL3 ratio. A)** FBN2 is downregulated in fibrotic tissue. B) COL1A1 is downregulated in fibrotic tissue C) COL1A2 is downregulated in fibrotic tissue D) COL3A1 is slightly downregulated in fibrotic tissue. For all above panels: Dark grey bars represent healthy tissue, light grey bars fibrotic tissue. Y-axes represent percentage of total protein detected in the decellularized tissue, p≥0.05. D) Ratio COL1/ COL3. For this ratio we used (LFQ(COL1A1)+LFQ(COL1A2))/LFQ(COL3A1) per condition. Dark grey bars represent healthy tissue, light grey bars fibrotic tissue. Y-axes represent ratio, although we see a trend of upregulated ratio, the result is not significant (p≥0.05).
(PDF)

**S1 File.**
(XLSX)

**S2 File.**
(XLSX)

**S1 Table. Antibodies used for immunohistochemical stainings.**
(PDF)

**S2 Table. Matrisome enrichment differs between SDS and Triton decellularization.**
(PDF)

**S3 Table. Differential identification of matrisome components in comparison between SDS and Triton decellularization.**
(PDF)

**S4 Table. Fold change of different matrisome components as depicted in the Volcano plot of Fig 4C.**
(PDF)

## Author Contributions

**Conceptualization:** Emma C. Linssen, Laetitia M. O. de Kort, Petra de Graaf.

**Data curation:** Emma C. Linssen, Jeroen Demmers, Roos van Dam, Maria Novella Nicese, Petra de Graaf.

**Formal analysis:** Emma C. Linssen, Jeroen Demmers, Christian G. M. van Dijk, Roos van Dam, Maria Novella Nicese, Caroline Cheng, Petra de Graaf.

**Investigation:** Emma C. Linssen, Maria Novella Nicese, Petra de Graaf.

**Methodology:** Jeroen Demmers, Christian G. M. van Dijk, Roos van Dam, Caroline Cheng, Laetitia M. O. de Kort, Petra de Graaf.

**Resources:** Laetitia M. O. de Kort.

**Software:** Jeroen Demmers.

**Supervision:** Jeroen Demmers, Laetitia M. O. de Kort, Petra de Graaf.

**Validation:** Jeroen Demmers, Petra de Graaf.

**Visualization:** Christian G. M. van Dijk, Maria Novella Nicese, Petra de Graaf.

**Writing – original draft:** Emma C. Linssen, Petra de Graaf.

**Writing – review & editing:** Emma C. Linssen, Jeroen Demmers, Christian G. M. van Dijk, Maria Novella Nicese, Caroline Cheng, Laetitia M. O. de Kort.

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
