## [Decision Letter · Decision Letter 0]

25 Jul 2023

PONE-D-23-17947Extracellular matrix analysis of fibrosis: a step towards tissue engineering for urethral stricture diseasePLOS ONE

Dear Dr. de Graaf,

Thank you for submitting your manuscript to PLOS ONE. After careful consideration, we feel that it has merit but does not fully meet PLOS ONE’s publication criteria as it currently stands. Therefore, we invite you to submit a revised version of the manuscript that addresses the points raised during the review process.

We look forward to receiving your revised manuscript.

Kind regards,

Panayiotis Maghsoudlou

Academic Editor

PLOS ONE

Journal Requirements:

Reviewers' comments:

Reviewer's Responses to Questions

**Comments to the Author**

1. Is the manuscript technically sound, and do the data support the conclusions?

Reviewer #1: Partly

Reviewer #2: Yes

Reviewer #3: Yes

Reviewer #4: Yes

2. Has the statistical analysis been performed appropriately and rigorously? 

Reviewer #1: N/A

Reviewer #2: Yes

Reviewer #3: Yes

Reviewer #4: Yes

3. Have the authors made all data underlying the findings in their manuscript fully available?

Reviewer #1: Yes

Reviewer #2: Yes

Reviewer #3: No

Reviewer #4: Yes

4. Is the manuscript presented in an intelligible fashion and written in standard English?

Reviewer #1: Yes

Reviewer #2: Yes

Reviewer #3: Yes

Reviewer #4: Yes

5. Review Comments to the Author

Reviewer #1: 1. In the current routine urethral reconstruction, urethra is only reconstructed with epithelial tissue, and the corpus spongiosum (CS) is not reconstructed. Literature on CS reconstruction is scarce. The author aims at the reconstruction of CS, which should be appreciated.

2. The reconstruction using tissue engineering technique should be based on normal cells, normal structure and normal extracellular matrix of normal healthy tissue in order to furfill the biomimetic reconstruction. Moreover, In real surgery, In real urethral stricture disease, there is fibrosis of the urethral epithelium and deeper layers including the CS. Urethral strictures can be best treated with open urethroplasty, where the fibrotic urethra and CS is completely excised, followed by primary anastomosis on the site of healthy tissue could be found. Therefore, if tissue engineering reconstruction of urethra is the goal of this research, the focus of research should be on the normal urethra, normal epithelia, normal CS and its normal extracellular matrix. The present study focused on the fibrotic CS and its extracellular matrix within the urethral stenosis, The information provided to us should primarily facilitate the comprehension of corpus spongiosum fibrosis formation mechanism, while being restricted for tissue engineering research on CS.

3. The author focuses on the proteomic comparison of normal spongiosum and fibrotic spongiosum in urethral stenosis, maybe he wants to provide strategies for the development of tissue engineering spongiosum to avoid the fibrosis formation after they are transplanted into body. If so, the fibrosis of tissue engineered constructs is mainly due to insufficient blood supply of the construct, lack of vascularization leads to fibrosis and graft rejection, which is completely different from the fibrosis of spongiosum caused by urethral stenosis. Furthermore, the authors' proteomic study of the extracellular matrix of urethral spongiosum only presents findings without exploring the specific roles of these altered proteins in urethral stricture and CS fibrosis. As a result, their work merely offers a comparison of phenomena rather than an investigation into underlying mechanisms of urethral stricture or potential applications for tissue engineering CS.

Reviewer #2: The authors presented a proteomic analysis of healthy vs diseased corpus spongiosum using LC-MS/MS. The corpus spongiosum is an important component of the male urethra which provides copious vascular supply and physical protection. It is interesting that the distrubtion of protein types were similar between each groups.

The largest limitation is the unclear history of both "healthy" controls, who are likely to have had hormonal exposure and unclear history of disease controls (lichen sclerosis v injury v prior urologic procedure v first stricture repair v hx of hypospadius). If known, please discuss the background of diseased specimens further. Discussion mentioned the limitation of using the transgender population as healthy controls, but should authors should discuss how hormones are known to effect corpous spongiosum (i.e. in hypospadiac infants) and if there is literature on hormone changes on the proteins that were up/downregulated.

Title:

--It would be helpful to include the methods (i.e. proteomics via LC-MS/MS) in the title

Abstract:

--clear, concise, interesting

Introduction:

--well written, no suggestions

Methods:

--Diseased tissues, did these patients have lichen sclerosis, injury, prior urologic procedure, prior repair or were these all first presenting urethral stricture disease?

--Control tissues were obtained from gender confirmation surgery. Limitations of this are briefly mentioned in the discussion.

--How was an n-number of 3 chosen?

Results:

--Tables/figures/graphs are clear, concise, with appropriate labels and error bars

Discussion:

--Good discussion of up and downregulated proteins in fibrotic tissues. Could discuss how hormone therapy may effect these proteins in particular to provide more rationale for why transgender tissue is sufficient in this analysis.

Reviewer #3: Authors study the extracellular matrix of the fibrous tissue in regions of excised penile urethral strictures and compare these to normal extracellular matrix from corpus spongiosum taken from transgender procedures. The authors demonstrate differences in subtypes of extracellular matrix materials between fibrotic and normal samples, which may aide augmenting urethral reconstructive surgery which currently only involves mucosal grafting. Authors postulate that deficient biomatrix materials may be bioengineered as a future aim for surgical improvement. While the study is limited in the number of samples (3 strictured and 3 “normal”) the analysis is well done and nicely validated. The discussion is excellent with well described function of deficient matrix proteins in other organ fibrotic conditions and a very nicely written paragraph discussing the limitations.

Major:

The tables cited in the text are not provided in the manuscript download. Supplemental tables are presented but not he primary Tables which would need to be reviewed in context with the manuscript.

Minor:

1. Methods: Please state in decellularization procedure whether you are working with frozen tissue or FFPE. I assume the former but it should be stated.

2. Results (line 190): inter-alfa-trypsin should be inter-alpha-trypsin

3. Results –“validation of ECM in Immunohistochemistry section”: It isn’t clear why you stained FBN1 and EMILIN1 when your results show differences in FBN2 and EMILIN2. I assume it is because of antibody availability for 1’s and not 2’s. Counld you please state why you took this approach.

4. Discussion (line 232, 243, 251): “higher expressed” consider rewording “more highly expressed”

5. Discussion (lines 243-253): In the discussion of POSTN – nice discussion of findings related to skin wound healing. You mention that POSTN is produced in pericytes. One difference between CS and skin is that the erectile tissue is rich in pericytes in the normal state. Some of the difference you describe may relate to loss of the pericytes in the process of “spongiofibrosis”.

Reviewer #4: Despite the burgeoning numbers of papers related to interstitial fibrosis of various organs during the past decade, many papers under report on the actual composition of the ECM scaffold in the fibrosing tissues. If in some cases, the changes in the nature and composition of the ECM matrix is reported, it may reflect a cursory investigation of relative mRNA abundance of the usual fibrillar collagens, including type I and type III alpha-monomers. While the matrix is, in some organs, dominated by these two fibrillar collagen species, the ECM is complex and a brief survey too often diminishes the role(s) of other protein components of the altered ECM. In the current paper, the authors sought to develop a tissue engineering strategy for alleviating issues found with the fibrosed urogenital tract, and point out that “it is necessary to know the protein composition of the corpus spongiosum”. While true, this approach requires the development of some innovative methodologies, or at least, an extension of those techniques. In the current work the authors analyzed the localization of ECM components in human healthy and fibrotic corpus spongiosum using immunohistological techniques. The overarching experimental design included an assessment of healthy normal tissues as well as fibrosed samples. In general the paper is very well written and the techniques used were adequate – further, the conclusions were within the confines of the data presented. The impact of the work could be improved if the following comments are given some attention.

Specific comments

1. Is is possible to include additional results eg, IF staining for collagen 4 alpha 1 within Figure 1? This would inject a modicum of novelty to the depiction of collagens I and III.

2. Matrisome proteins depicted in the volcano plot of Figure 3 is one of the best Figures of the paper, and provides some specific and interesting data. Again it appears to represent the tip of the iceberg so to speak. Is it possible to select two or three proteins from within each generalized group of matrisome protein classes to highlight these results in a separate panel? The ECM associated proteins and “regulators” are not well represented in the current literature.

3. Figure 5 shows elastin staining. Elastin is notoriously slow to degrade and is almost never added to tissues. Alternatively, elastin fenestration is a dynamic process that is ongoing with ageing tissues and may be altered in fibrosis. Is there any data to interrogate the status of elastase activities in CS samples (healthy and fibrosed)? Please comment.

6. PLOS authors have the option to publish the peer review history of their article (what does this mean?). If published, this will include your full peer review and any attached files.

Reviewer #1: **Yes: **Fang Chen

Reviewer #2: **Yes: **Nora M Haney

Reviewer #3: No

Reviewer #4: No

---

## [Author Response · Author response to Decision Letter 0]

10 Nov 2023

We have uploaded our response to reviewers as a document in the attached files

---

## [Editor Report · Decision Letter 1]

13 Nov 2023

Extracellular matrix analysis of fibrosis: a step towards tissue engineering for urethral stricture disease

PONE-D-23-17947R1

Dear Dr. de Graaf,

We’re pleased to inform you that your manuscript has been judged scientifically suitable for publication and will be formally accepted for publication once it meets all outstanding technical requirements.

Kind regards,

Panayiotis Maghsoudlou

Academic Editor

PLOS ONE

---

## [Editor Report · Acceptance letter]

20 Nov 2023

PONE-D-23-17947R1 

Extracellular matrix analysis of fibrosis: a step towards tissue engineering for urethral stricture disease 

Dear Dr. de Graaf:

I'm pleased to inform you that your manuscript has been deemed suitable for publication in PLOS ONE. Congratulations! Your manuscript is now with our production department. 

Kind regards, 

on behalf of

Dr. Panayiotis Maghsoudlou 

Academic Editor

PLOS ONE